# A retrospective study of vector borne disease prevalence among anemic dogs in North Carolina

**Katie L. Anderson[1], Adam Birkenheuer[1], George E. Moore[2], Allison Kendall[1]***

**1** Department of Veterinary Clinical Sciences, NC State University College of Veterinary Medicine, Raleigh, NC, United States of America, **2** Department of Veterinary Administration, Purdue University College of Veterinary Medicine, West Lafayette, IN, United States of America

* arkendal@ncsu.edu

**Data Availability Statement:** All relevant data are within the paper and its Supporting Information files.

**Funding:** The authors received no specific funding for this work.

## Abstract

### Background

Anemia is an important cause of morbidity and mortality in dogs. Further understanding of the prevalence of vector borne diseases (VBD) in anemic dogs is needed.

### Objectives

The objective of this retrospective study was to describe the rate of exposure to or infection with VBD among anemic dogs presented to a teaching hospital in North Carolina and to further characterize the anemia in dogs with VBD exposure.

### Animals

A total of 597 anemic dogs that were concurrently tested for VBD were examined at a referral veterinary hospital between January 2012 and December 2018.

### Methods

Retrospective descriptive study. Demographic, clinicopathologic, and VBD testing data were obtained from medical records.

### Results

Of the 597 anemic dogs examined, 180 (30.15%; 95% CI: 26.49–34.01%) tested positive for one or more VBD. There was no difference in the severity of anemia or the proportion of dogs displaying a regenerative anemia between dogs testing positive and negative for VBD.

### Conclusions

A large proportion of anemic dogs from this region test positive for exposure to or infection with VBD. Our study supported the use of PCR and serology run in parallel to maximize the chance of detecting exposure to or infection with VBD compared to either serology or PCR alone. At this time, it is unknown whether infection with VBD contributed to the development

**Competing interests:** The authors have declared that no competing interests exist.

of anemia in these patients. However, given the prevalence of VBD exposure in anemic dogs, testing for VBD in anemic patients from this region of the United States is warranted.

## Introduction

Anemia is one of the most common hematologic abnormalities identified in dogs and is frequently a marker of underlying disease [1]. Anemia results from three main pathophysiologic mechanisms: hemorrhage, hemolysis, and decreased production [1–3]. These mechanisms may be induced by a variety of disease conditions, including several common vector borne diseases [1]. Identifying the underlying cause of anemia not only facilitates appropriate treatment but can improve the prognosis and survival of veterinary patients.

Previous retrospective studies have examined the relative frequency of various causes of anemia in dogs [4–7]. However, only a few studies have examined infectious causes of anemia, such as vector borne diseases. In one study, unspecified infectious disease was considered to be the cause of anemia in 35/456 (7.7%) dogs with no evidence of blood loss [6]. In Taiwan, 287/2037 (11%) dogs developed anemia secondary to infectious disease, with *Babesia gibsoni* being the most commonly diagnosed pathogen in severely anemic dogs (hematocrit <13%) [7]. In France, vector borne diseases were associated with 17.2% of anemic cases and were the second-most common cause of hemolysis. *Babesia* species and *Mycoplasma* species were the most commonly identified infectious agents and were found in 8.2% and 7.5% of anemic dogs, respectively [8]. These studies suggest that vector borne disease may be an important cause of anemia in dogs.

Vector borne infections are prevalent in the Southeastern United States. Of 118 dogs tested in North Carolina from 2009–2013, 97 dogs (82%) had been exposed to or were actively infected with one or more vector borne pathogens [9]. One study suggested that using a combination of serology and PCR in parallel improved detection of dogs that were exposed to or infected with VBD [10]. However, to the authors' knowledge, no study has reported the use of serology and PCR run in parallel in a large number of anemic dogs. Therefore, the purpose of this study was to describe the prevalence of exposure to or infection with vector borne diseases among anemic dogs presented to a teaching hospital in North Carolina. The first aim was to determine the proportion of anemic dogs exposed to or infected with each vector borne disease of interest, including *Babesia* spp., *Ehrlichia canis*, spotted-fever group *Rickettsia*, *Bartonella* spp., *Anaplasma* spp., *Mycoplasma* spp., and *Borrelia burgdorferi*. Our second aim was to compare the results of serologic and molecular testing in this patient population. The third aim was to further characterize the anemia associated with vector borne disease in terms of regeneration or characteristics of immune mediated hemolytic anemia, including spherocytosis, a positive saline agglutination test, and/or a positive Coombs test. The final aim was to describe the proportion of dogs that had additional hematologic abnormalities, including thrombocytopenia, neutropenia or pancytopenia.

## Materials and methods

### Case selection criteria

The hematology database at the NC State University (NCSU) Veterinary Hospital was searched retrospectively for dogs with a complete blood count (CBC) and/or a reticulocyte count performed at this hospital from January 2012 to December 2018. Anemic patients, as defined by a packed cell volume (PCV) of less than 30%, were identified on CBC. These

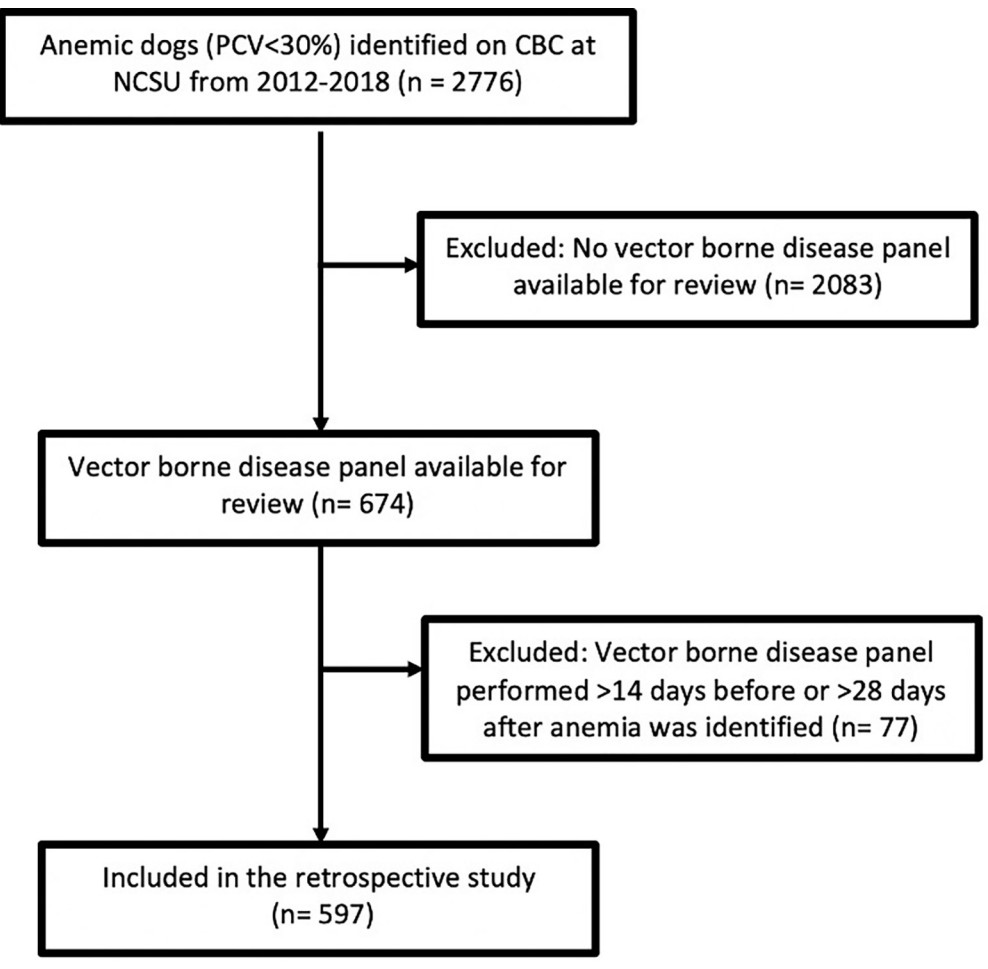

**Fig 1. Flowchart describing inclusion and exclusion criteria for the study.**

patients were then cross-referenced with the NCSU Vector Borne Disease Diagnostic Laboratory (VBDDL) database to identify anemic dogs that had ≥1 canine vector borne disease test performed. In an effort to include tests that were likely to be associated with the episode of anemia, only vector borne disease (VBD) tests performed within 14 days before or 28 days after anemia was first identified on a CBC were included (Fig 1).

## Medical records

When available, the following demographic information was extracted from the medical records of the dogs included in the study: age at the time of presentation, sex, breed, weight, and home state.

Information collected from the initial CBC on which anemia was identified included: red blood cell (RBC) count, HCT, packed cell volume (PCV), total solids, mean corpuscular volume (MCV), mean corpuscular hemoglobin concentration (MCHC), reticulocyte count, presence of agglutination, presence of organisms on the blood smear, white blood cell count, segmented neutrophil count, band neutrophil count, and platelet count. Regenerative anemia was defined as a reticulocyte count of greater than or equal to 60,000 cells/uL [2]. Neutropenia was defined as less than 2,841 cells/uL, and thrombocytopenia was defined as less than 190,000

cell/uL based on the institutional reference intervals. The following RBC morphologic indices were also recorded on a standardized scale from 0 (not present) to 5 (marked): nucleated RBC, spherocytes, anisocytosis, poikilocytosis, polychromasia, acanthocytes, keratocytes, echino-cytes, ghost cells, Heinz bodies, Howell Jolly bodies, macrocytosis, microcytosis, schistocytes, stomatocytes, and target cells. The presence or absence of microorganisms within erythrocytes was recorded. The results of a Coombs test were also collected, if performed. Finally, the results of any VBD serology and polymerase chain reaction (PCR) testing performed were collected.

### Vector borne disease testing

Samples were submitted by the attending clinician to the NCSU VBDDL for individual sero-logic or polymerase chain reaction (PCR) tests or for a comprehensive canine vector borne pathogen panel. Over the six-year study period, a number of samples were not tested for all pathogens based on clinician discretion, changes in pathogen testing capabilities at the NCSU VBDDL, or both.

Serum samples were tested by indirect IFA for *Babesia canis*, *Babesia gibsoni*, *Ehrlichia canis*, *Rickettsia* spp., *Bartonella vinsonii*, *Bartonella henselae*, and *Bartonella koehlerae* and/or were tested for *Borrelia burgdorferi*, *Ehrlichia canis*, *Ehrlichia ewingii*, *Anaplasma phagocyto-philum*, and *Anaplasma platys* antibodies and *Dirofilaria immitis* antigen using a commercial ELISA-based assay (SNAP 4DX or SNAP 4DX Plus, IDEXX Laboratories Inc, Westbrooke, Maine). All NCSU VBDDL IFA antigens were grown *in vitro* or *in vivo* by personnel in the VBDDL using bacterial or protozoan strains of canine or feline origin. A positive IFA sample was defined as having an IFA titer ≥1:64. By IFA testing, serologic cross-reactivity occurs between spotted fever group rickettsiae, including both pathogenic and non-pathogenic strains. Serologic cross-reactivity also occurs between *Babesia gibsoni* IFA and *Babsia canis* IFA. With SNAP 4Dx Plus assays, cross-reactivity is known to occur between *Anaplasma pha-gocytophilum* and *Anaplasma platys* and the *Ehrlichia* spp. positive spot detects antibodies against both *Ehrlichia canis* and *Ehrlichia ewingii*.

Samples were tested by polymerase chain reaction (PCR) for the following pathogens: *Ana-plasma* spp., *Babesia* spp., *Bartonella* spp., *Ehrlichia* spp., hemotropic *Mycoplasma* spp., and *Rickettsia* spp. Primers, PCR conditions, and negative and positive controls were used as previ-ously described [11–22]. Briefly, DNA was extracted from stored canine ethylenediaminetetra-acetic acid (EDTA) whole blood samples using a commercially available MagAttract DNA blood kit (Qiagen Inc, Chatsworth, CA) according to the manufacturer's instructions. For all samples, the DNA concentration was quantified by spectrophotometry, and the absence of PCR inhibitors was demonstrated by amplification of a fragment of the glyceraldehyde-3-phosphatase dehydrogenase gene. PCR screening was performed as previously described [11–22]; a comprehensive list of the PCR primers, reaction volumes, and reaction conditions used is available in S1 Table. Products were analyzed by 2% agarose gel electrophoresis con-taining 0.2ug ethidium bromide/mL under ultraviolet light. Canine DNA from a healthy sub-ject was used as a PCR negative control. Positive controls are listed in S1 Table. A comprehensive vector borne pathogen panel was defined as a panel of PCR tests (all genera described above), immunofluorescent assays (IFA) tests (all assays described above with possi-ble exception of *B. gibsoni* IFA which was not available throughout the entire study period), and either SNAP 4Dx or SNAP 4Dx Plus (IDEXX Laboratories Inc, Westbrooke, Maine).

### Statistical analysis

Nominal or categorical data was summarized as proportions or percentages. Summary statis-tics for numerical data, e.g. age, weight, and PCV, were expressed as median (range).

### Ethics statement

This manuscript was retrospective in nature and utilized data from clinical patients. All data was anonymized prior to use. Given the retrospective nature of this manuscript, prospective approval from the NC State Institution Animal Care and Use Committee (IACUC) was not necessary and a formal waiver of ethics approval was not obtained.

## Results

### Characteristics of the study population

During the 2012–2018 study period, a total of 2776 anemic dogs were identified on CBC at our institution. Of these, 597 anemic dogs (21.5%) met the inclusion criteria (Fig 1). Individual patient information can be found in S2 Table. A total of 112 different dog breeds were included, with Labrador Retrievers (10.72%), mixed breed dogs (7.27%), American Cocker Spaniels (3.69%), and Golden Retrievers (3.69%) being the most common. Dogs ranged in age from 2 months to 17 years, with a median of 7.62 years. The median body weight was 16.8 kg (range: 1.2–79.0 kg). There were 314 spayed females (52.6%), 218 neutered males (36.52%), 50 intact males (8.38%), and 15 intact females (2.51%).

### Results of vector borne disease testing

Of the 597 anemic dogs included in the study, 181 (30.3%; 95% CI: 26.77–34.12%) tested positive for VBD (Fig 2). Of these 181 dogs, 132 dogs (22.1%) tested positive for one VBD and 49 dogs (8.2%) tested positive for more than one VBD (S3 Table).

Of the 597 dogs included, a comprehensive VBD panel was performed for 474 dogs. The remaining 123 dogs had selected tests performed at the clinician's discretion. As each dog had multiple tests performed, a total of 8,977 serology or PCR tests were analyzed, of which 319 tests (3.55%) were positive for infection with or exposure to VBD (Table 1).

In six patients, microorganisms were observed by microscopy on a blood smear created from a venous sample. Three dogs had intra-erythrocytic inclusions consistent with *Babesia*

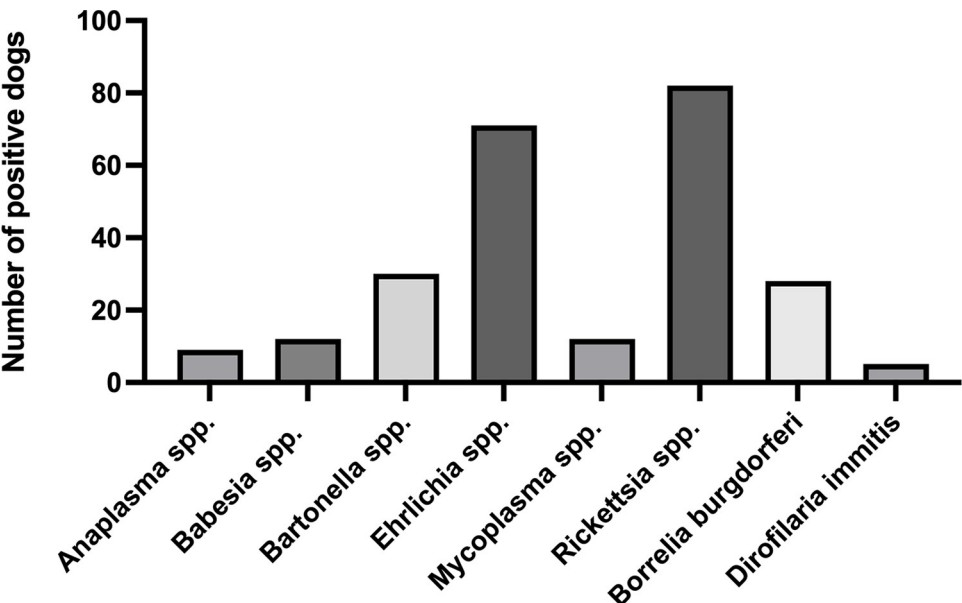

**Fig 2. Number of dogs testing positive for exposure to or infection with individual vector borne disease.**

**Table 1. Results of individual vector borne disease (VBD) tests in anemic dogs during the study period.**

| Test performed | Number of positive tests | Total number of tests performed | Percent positive (%) |
|---|---|---|---|
| Any VBD | 319 | 8,977 | 3.55 |
| *Anaplasma* PCR | 4 | 478 | 0.84 |
| *Anaplasma* SNAP | 6 | 573 | 1.05 |
| *Babesia* spp. PCR | 8 | 511 | 1.57 |
| *Babesia canis* IFA | 7 | 556 | 1.26 |
| *Babesia gibsoni* IFA | 8 | 457 | 1.75 |
| *Bartonella* spp. PCR | 0 | 485 | 0 |
| *Bartonella vinsonii* IFA | 20 | 563 | 3.55 |
| *Bartonella hensalae* IFA | 21 | 563 | 3.73 |
| *Bartonella koehlerae* IFA | 21 | 513 | 4.09 |
| *Ehrlichia* spp. PCR | 11 | 489 | 2.25 |
| *Ehrlichia* spp. SNAP | 67 | 573 | 11.69 |
| *Ehrlichia canis* IFA | 19 | 555 | 3.42 |
| *Mycoplasma* PCR | 12 | 478 | 2.51 |
| *Rickettsia* spp. PCR | 0 | 480 | 0 |
| *Rickettsia* spp. IFA | 82 | 557 | 14.72 |
| *Borrelia burgdorferi* SNAP | 28 | 573 | 4.89 |
| *Dirofilaria immitis* SNAP | 5 | 573 | 0.87 |

Table abbreviations: VBD, vector borne disease; PCR, polymerase chain reaction; IFA, immunofluorescence assay; spp., species.

spp. Two patients had visible small form organisms. Both patients tested PCR positive for *Babesia gibsoni*. One of them also tested positive for both *Babesia canis* and *Babesia gibsoni* by IFA; IFA testing was not performed in the other patient. One patient had visible large form organisms. PCR testing was not performed in this patient, and the patient tested negative for both *B. canis* and *B. gibsoni* by IFA. One patient had visible morulae within neutrophils; this patient tested negative for both *Ehrlichia* and *Anaplasma* by PCR and negative for *Anaplasma* by SNAP, but positive for *Ehrlichia* by both SNAP and IFA. Finally, two patients had visible microfilariae, both of which tested SNAP positive for *D. immitis*.

## Comparison of testing methods

For multiple vector borne diseases, both PCR and serology testing (SNAP 4Dx Plus or IFA) were performed on the same sample, and the agreement between these testing methods was examined. The agreement between PCR and serology varied greatly depending on the infectious agent (range: 0–36.4% agreement between serology and PCR). For *B. canis*., 36.4% of dogs tested positive by both PCR and serology, and for *B. gibsoni*, 28.6% of dogs tested positive by both PCR and serology. For *Bartonella* spp. and *Rickettsia* spp., none of the dogs tested positive by both PCR and serology (Table 2). Of the 474 dogs that had a comprehensive VBD panel, 157 (33.12%) had at least one positive test. If only PCR assay results had been considered, 32 (6.75%) of these dogs would have tested positive and no co-infections would have been detected. If only serologic assay results had been considered, a total of 147 (31.01%) of these dogs would have tested positive, with 44 (9.28%) dogs having evidence of co-exposure to 2 or more VBD and 123 (25.95%) testing positive for exposure to a single VBD. Despite detecting a greater number of dogs exposed to or infected with VBD, serology alone would have missed infections in 2 dogs infected with *Anaplasma* spp., 2 dogs infected with *Babesia* spp., 1 dog with *Ehrlichia* spp. and 12 dogs with *Mycoplasma* spp. (please note that there are no serologic assays for *Mycoplasma* spp.).

**Table 2. Agreement between PCR and serology test results for dogs that had both tests and had at least one positive test.**

| Disease (tests compared) | Number of positive dogs | Positive by PCR only (%) | Positive by serology only (%) | Positive by PCR and serology (%) |
|---|---|---|---|---|
| *Anaplasma* spp. (PCR and SNAP 4Dx Plus) | 9 | 3 (33.3%) | 5 (55.6%) | 1 (11.1%) |
| *Babesia* spp. (PCR and *Babesia canis* IFA) | 10 | 3 (30%) | 3 (30%) | 4 (40%) |
| *Babesia* spp. (PCR and *Babesia gibsoni* IFA) | 10 | 2 (20%) | 3 (30%) | 5 (50%) |
| *Bartonella* spp. (PCR and *Bartonella vinsonii* IFA) | 20 | 0 (0%) | 20 (100%) | 0 (0%) |
| *Bartonella* spp. (PCR and *Bartonella hensalae* IFA) | 21 | 0 (0%) | 21 (100%) | 0 (0%) |
| *Bartonella* spp. (PCR and *Bartonella koehlerae* IFA) | 21 | 0 (0%) | 21 (100%) | 0 (0%) |
| *Ehrlichia* spp. (PCR and SNAP 4Dx Plus) | 69 | 2 (2.9%) | 58 (84.1%) | 9 (13.04%) |
| *Ehrlichia* spp. (PCR and *Ehrlichia canis* IFA) | 23 | 4 (17.4%) | 12 (52.2%) | 7 (30.4%) |
| *Rickettsia* spp. (PCR and IFA) | 82 | 0 (0%) | 82 (100%) | 0 (0%) |

Table abbreviations: PCR, polymerase chain reaction; IFA, immunofluorescence assay.

## Characterization of anemia

The range and median packed cell volume (PCV) for anemic dogs testing positive and negative for VBD is presented in S1 Fig. The severity of anemia, as measured by PCV, varied widely among dogs testing positive for each individual VBD (Fig 3).

Four-hundred and eighty-nine dogs had a reticulocyte count performed. Two-hundred and eighty-eight dogs had reticulocyte counts < 60,000/µl and 201 dogs had

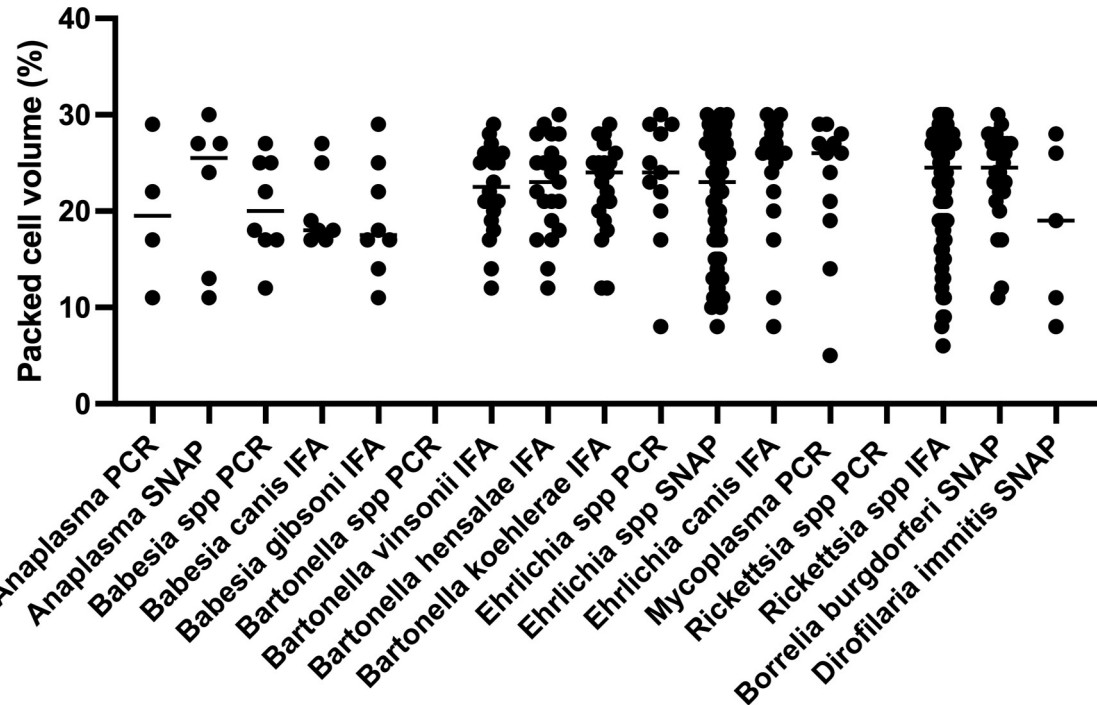

**Fig 3. Distribution and frequency of packed cell volume in dogs testing positive for individual vector borne disease (VBD).** The packed cell volume (y axis) of dogs testing positive each VBD test (x axis) is shown. Each dot represents an individual dog. The bar represents the median packed cell volume for each group. PCR, polymerase chain reaction; IFA, immunofluorescence assay.

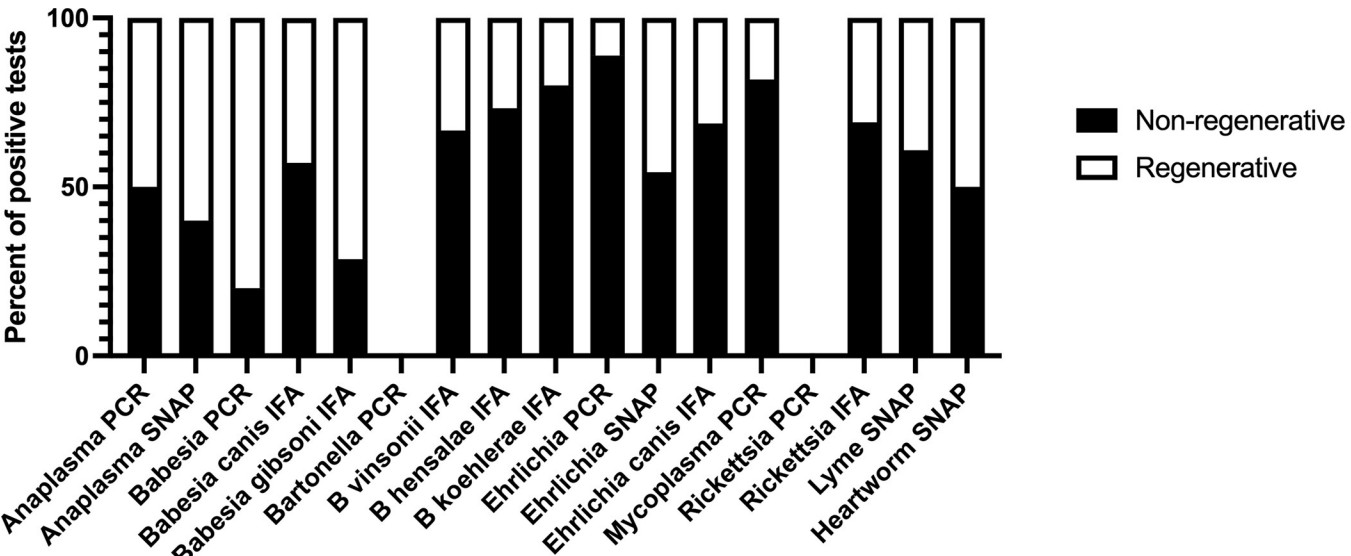

**Fig 4. Presence of regeneration in dogs testing positive for individual vector borne diseases.** A regenerative anemia was defined as a reticulocyte count greater than 60,000 cells/uL. Individual vector borne disease tests are displayed on the x-axis, and the percent of positive tests is displayed on the y-axis. The black bars represent a non-regenerative anemia (reticulocyte count <60,000 cells/uL), and the white bars represent a regenerative anemia (reticulocyte count ≥60,000 cells/uL). PCR, polymerase chain reaction; IFA, immunofluorescence assay.

reticulocytes ≥ 60,000/μl. Of the dogs testing positive for VBD, 60.1% (92/151) displayed a non-regenerative anemia, while 39.1% (59/151) displayed a regenerative anemia. Of dogs testing negative for VBD, 57.9% (196/338) displayed a non-regenerative anemia and 42.0% (142/338) displayed a regenerative anemia (S2 Fig). Dogs infected with certain individual vector borne diseases displayed a higher proportion of regenerative or non-regenerative anemia (Fig 4). Of the dogs that had reticulocyte counts reported that also tested PCR positive for *Babesia* spp., 4/5 (80%) displayed a regenerative anemia. Of the dogs that had reticulocyte counts reported that also tested positive for *Ehrlichia* spp. 8/9 (88.9%) had a non-regenerative anemia. Of the dogs that had reticulocyte counts reported that also tested positive for *Mycoplasma* spp. by PCR 9/11 (81.8%) displayed a non-regenerative anemia.

Dogs testing positive and negative for VBD were also evaluated for characteristics of immune mediated hemolytic anemia, including spherocytosis, agglutination, and/or a positive Coombs test (S4 Table) [14]. The presence of spherocytes, agglutination, and/or a positive Coombs test varied among individual vector borne diseases (Table 3). Overall, 11.4% of dogs (68/597) demonstrated two or more signs of immune mediated destruction that we evaluated in this study (spherocytosis, agglutination and a positive Coombs test). Of these dogs, 13.2% (9/68) tested positive for exposure to or infection with one or more VBD: two dogs were positive for *Ehrlichia* by PCR only, 1 dog was positive for *Anaplasma* by PCR only, 2 dogs were positive for both *Ehrlichia* by PCR and *Rickettsia* by IFA, 2 dogs were positive for *Rickettsia* by IFA only, 1 dog was positive for *Ehrlichia* by IFA only, 1 dog was positive for *Ehrlichia* by SNAP only.

## Presence of thrombocytopenia, neutropenia, or pancytopenia in dogs with and without VBD

The presence of thrombocytopenia, neutropenia and/or pancytopenia varied among anemic dogs testing positive for VBD (Table 4). Of the dogs testing positive for *Babesia* spp., 100% (8/8) of PCR positive dogs, 85.7% (6/7) of *B. canis* IFA positive dogs, and 75% (6/8) of *B. gibsoni*

**Table 3. Presence of immune mediated hemolytic anemia markers in dogs testing positive for individual vector borne diseases.**

| Test performed | Number of positive tests with spherocytes present /total number of positive tests examined | Number of tests with agglutination present / total number of positive tests examined | Number of positive Coombs tests/total number of positive tests examined |
|---|---|---|---|
| *Anaplasma* PCR | 0/4 | 0/4 | 0/0 |
| *Anaplasma* SNAP | 2/6 | 1/6 | 0/1 |
| *Babesia* spp. PCR | 0/8 | 0/8 | 0/1 |
| *Babesia canis* IFA | 0/7 | 1/7 | 0/1 |
| *Babesia gibsoni* IFA | 1/8 | 1/8 | 0/1 |
| *Bartonella* spp. PCR | 0/0 | 0/0 | 0/0 |
| *Bartonella vinsonii* IFA | 1/20 | 2/20 | 1/2 |
| *Bartonella henselae* IFA | 1/21 | 0/21 | 0/1 |
| *Bartonella koehlerae* IFA | 1/21 | 0/21 | 0/3 |
| *Ehrlichia* spp. PCR | 0/11 | 0/11 | 1/2 |
| *Ehrlichia* spp. SNAP | 11/67 | 9/67 | 6/15 |
| *Ehrlichia canis* IFA | 1/19 | 1/19 | 1/2 |
| *Mycoplasma* PCR | 0/12 | 2/12 | 0/0 |
| *Rickettsia* spp. PCR | 0/0 | 0/0 | 0/0 |
| *Rickettsia* spp. IFA | 9/82 | 11/82 | 5/12 |
| *Borrelia burgdorferi* SNAP | 1/28 | 1/28 | 1/5 |
| *Dirofilaria immits* SNAP | 0/5 | 0/5 | 0/1 |

Table abbreviations: IMHA, immune mediated hemolytic anemia; PCR, polymerase chain reaction; IFA, immunofluorescence assay; spp, species.

IFA positive dogs were thrombocytopenic. All dogs testing positive for *Anaplasma* spp. by SNAP 4Dx Plus were also thrombocytopenic (6/6). Of the dogs testing PCR positive for *Mycoplasma* spp., 41.7% (5/12) were thrombocytopenic. Dogs testing positive for *Ehrlichia* spp. by PCR or IFA had higher rates of neutropenia (2/11 (18.2%) of PCR positive dogs; 5/18 (27.8%) of IFA positive dogs) and pancytopenia (2/11 (18.2%) of PCR positive dogs; 3/18 (16.8%) of IFA positive dogs) compared to dogs that tested negative for VBD, of which 30/416 (7.21%) were neutropenic and 27/398 (6.8%) were pancytopenic.

## Discussion

The results of our retrospective study demonstrate that a large proportion of anemic dogs examined and treated at our institution have exposure to or infection with vector borne diseases (VBD). Of the nearly 600 anemic dogs included in our study, approximately 30% of these dogs were seropositive or PCR positive for one or more VBD. The prevalence of VBD among anemic dogs in our study was similar to 134 anemic dogs in France, of which 17.2% of anemic dogs tested positive for VBD [8]. Other studies examining the proportion of dogs in the Southeastern United States that are infected with or exposed to VBD have also found a similar prevalence rate [14,23].

Our study confirmed that the use of serology and PCR in parallel detected a larger number of dogs exposed to or infected with VBD than either methodology alone. A positive PCR assay,

**Table 4. Presence of thrombocytopenia, neutropenia, or pancytopenia in dogs testing positive for vector borne diseases.**

| Test performed | Number of thrombocytopenic animals/total number of positive tests | Number of neutropenic animals/total number of positive tests | Number of pancytopenic animals/total number of positive tests |
|---|---|---|---|
| *Anaplasma* PCR | 3/4 | 0/4 | 0/4 |
| *Anaplasma* SNAP | 6/6 | 0/6 | 0/6 |
| *Babesia* spp PCR | 8/8 | 1/8 | 1/8 |
| *Babesia canis* IFA | 6/7 | 0/7 | 0/7 |
| *Babesia gibsoni* IFA | 6/8 | 0/8 | 0/8 |
| *Bartonella* spp. PCR | 0/0 | 0/0 | 0/0 |
| *Bartonella vinsonii* IFA | 12/20 | 1/20 | 0/20 |
| *Bartonella hensalae* IFA | 14/21 | 1/21 | 0/21 |
| *Bartonella koehlerae* IFA | 13/21 | 2/21 | 1/21 |
| *Ehrlichia* spp. PCR | 6/11 | 2/11 | 2/11 |
| *Ehrlichia* spp. SNAP | 42/66 | 7/67 | 6/66 |
| *Ehrlichia canis* IFA | 8/18 | 5/19 | 3/18 |
| *Mycoplasma* PCR | 5/12 | 0/12 | 0/12 |
| *Rickettsia* spp. PCR | 0/0 | 0/0 | 0/0 |
| *Rickettsia* spp. IFA | 48/79 | 6/82 | 4/79 |
| *Borrelia burgdorferi* SNAP | 18/27 | 2/28 | 1/28 |
| *Dirofilaria immits* SNAP | 3/5 | 1/5 | 0/5 |

Table abbreviations: PCR, polymerase chain reaction; IFA, immunofluorescence assay; spp, species.

performed with the appropriate controls, provides nearly definitive evidence of a current infection, while a positive serologic assay provides circumstantial evidence that a current infection may be present. However, neither test is sufficient to determine whether or not a VBD contributed to the development of anemia. Due to the retrospective nature of this study, where testing, treatments and follow-up were not controlled, we did not attempt to assess causality. Our study will help clinicians in the region to establish appropriate expectations regarding the number of positive tests they could expect when testing dogs with anemia for VBD. In patients testing seropositive for VBD, careful consideration should be given as to whether treatment for VBD is indicated, and the clinical response to such treatments should be closely assessed. However, given the prevalence of VBD in anemic patients from the Southeastern United States, testing is likely warranted in this patient population.

In addition to evaluating the prevalence of VBD in anemic dogs, our study aimed to characterize the anemia associated with vector borne disease in terms of regeneration or characteristics of immune mediated hemolytic anemia, including spherocytosis, a positive saline agglutination test, and/or a positive Coombs test. Based on our findings, the severity or type of anemia (regenerative or non-regenerative) cannot be used to predict whether anemic dogs will test positive for any VBD. While dogs with certain infections had relatively higher proportions of regenerative or non-regenerative anemia, no infection was exclusively detected in dogs with either regenerative or non-regenerative anemia. Likewise, the presence of spherocytes, agglutination or positive Coombs test were not specific to dogs with or without exposure to or infection with VBD. Finally, the presence of concurrent thrombocytopenia, neutropenia or both did not appear to predict whether or not anemic dogs were exposed to or infected with VBD.

This study was retrospective in nature and was therefore subject to the limitations inherent in any retrospective analysis. There was no standardization of data collection at the time the patient presented, and minor variation in laboratory techniques may have occurred over the study period. As this is also a descriptive study, a control group has not been included. Clinicians elected to perform VBD testing based on the patient's clinical presentation or clinicopathologic results, which may have introduced bias into the patient selection process. While many anemic dogs were not tested for VBD during this time period, our study included anemic dogs presenting to all small animal hospital sections. Therefore, only some of the dogs underwent a diagnostic workup for causes of anemia, including VBD testing. In addition, not all dogs were tested for all VBD based on clinician discretion or changes in pathogen testing capabilities. All data collected from this study was obtained from patients presenting to a tertiary referral center and therefore may not reflect the first-opinion practices. Due to regional variations among tick species and disease prevalence, the results of our study may not be extrapolated to all regions of the United States. Establishing a direct cause and effect relationship between anemia and vector borne diseases will require controlled, epidemiologic and experimental studies. However, the information derived from our study could improve our current understanding of VBD exposure in anemic dogs.

Overall, our results provide a realistic expectation of the proportion of anemic dogs from the southeastern region of the United States that will test positive for VBD. It is appropriate to expect that the majority of anemic dogs will not be diagnosed with a VBD. However, VBD are identified in a subset of these dogs, and a diagnosis of a VBD could change the patient's prognosis, treatment, and ultimately, their outcome. In addition, we have demonstrated that specific characteristics of anemia, including regeneration, agglutination, spherocytosis, or a positive Coombs test, cannot predict infection with or exposure to VBD. Therefore, consideration should be given to testing all anemic dogs from this region for these diseases.

## Supporting information

**S1 Fig. Distribution and frequency of packed cell volume in dogs with and without vector borne disease (VBD).** The packed cell volume (y axis) of dogs testing positive or negative for VBD (x axis) is shown. Each dot represents an individual dog. The bar represents the median packed cell volume for each group (test positive median PCV 23% (range 5–30%); test negative median PCV 24% (range 4–30)).
(TIFF)

**S2 Fig. Number of dogs with regenerative or non-regenerative anemia that tested positive or negative for one or more vector borne disease (VBD).** Regeneration was defined as a reticulocyte count greater than or equal to 60,000 cells/uL (x-axis). The number of dogs in each group is shown on the y-axis. The black bars represent dogs testing negative for all VBD while the white bars represent dogs testing positive for one or more VBD.
(TIFF)

**S1 Table. PCR primers, reaction volumes, and reaction conditions.**
(XLSX)

**S2 Table. Individual patient information.**
(XLSX)

**S3 Table. Number of dogs testing positive for exposure to or infection with multiple vector borne diseases.**
(DOCX)

**S4 Table. Number of dogs with spherocytosis, agglutination, or a positive Coombs test that tested positive or negative for one or more vector borne disease (VBD).**
(DOCX)

## Acknowledgments

The authors would like to acknowledge Frances Torres Otero, Hayley Stratton, Dylan DeProspero, Angelle Abatte, Juliana Mills, and Stephen Gregory for their assistance with data collection.

## Author Contributions

**Conceptualization:** Katie L. Anderson, Adam Birkenheuer, Allison Kendall.

**Data curation:** Katie L. Anderson, Adam Birkenheuer, Allison Kendall.

**Formal analysis:** Katie L. Anderson, Adam Birkenheuer, George E. Moore, Allison Kendall.

**Funding acquisition:** Allison Kendall.

**Investigation:** Katie L. Anderson, Adam Birkenheuer, Allison Kendall.

**Methodology:** Katie L. Anderson, Adam Birkenheuer, George E. Moore, Allison Kendall.

**Project administration:** Katie L. Anderson, Adam Birkenheuer.

**Resources:** Katie L. Anderson, Adam Birkenheuer.

**Software:** Katie L. Anderson.

**Supervision:** Adam Birkenheuer, Allison Kendall.

**Validation:** Katie L. Anderson, Adam Birkenheuer.

**Visualization:** Katie L. Anderson, Adam Birkenheuer.

**Writing – original draft:** Katie L. Anderson.

**Writing – review & editing:** Katie L. Anderson, Adam Birkenheuer, Allison Kendall.

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
