## [Decision Letter · Decision Letter 0]

16 Aug 2023

PONE-D-23-14843A retrospective study of vector borne disease prevalence among anemic dogs in North Carolina.PLOS ONE

Dear Dr. Kendall,

Thank you for submitting your manuscript to PLOS ONE. After careful consideration, we feel that it has merit but does not fully meet PLOS ONE’s publication criteria as it currently stands. Therefore, we invite you to submit a revised version of the manuscript that addresses the points raised during the review process.

Please submit your revised manuscript by Sep 30 2023 11:59PM. If you will need more time than this to complete your revisions, please reply to this message or contact the journal office at plosone@plos.org. Please include the following items when submitting your revised manuscript:A rebuttal letter that responds to each point raised by the academic editor and reviewer(s). You should upload this letter as a separate file labeled 'Response to Reviewers'.A marked-up copy of your manuscript that highlights changes made to the original version. You should upload this as a separate file labeled 'Revised Manuscript with Track Changes'.An unmarked version of your revised paper without tracked changes. You should upload this as a separate file labeled 'Manuscript'.

We look forward to receiving your revised manuscript.

Kind regards,

Joshua Kamani, PhD

Academic Editor

PLOS ONE

Journal Requirements:

2. Thank you for including your ethics statement:  "Due to the retrospective nature of this manuscript, no IACUC approval was needed.".   

To comply with PLOS ONE submissions requirements, please provide the following information in the Methods section of the manuscript and in the “Ethics Statement” field of the submission form (via “Edit Submission”):  

*  Please indicate whether an animal research ethics committee prospectively approved this research or granted a formal waiver of ethics approval.*  Please enter the name of your Institutional Animal Care and Use Committee (IACUC) or other relevant ethics board. Also include an approval number if one was obtained.

*   If anesthesia, euthanasia, or any kind of animal sacrifice is part of the study, please include briefly in your statement which substances and/or methods were applied.

For additional information about PLOS ONE submissions requirements for ethics oversight of animal work, please refer to http://journals.plos.org/plosone/s/submission-guidelines#loc-animal-research  

Reviewers' comments:

Reviewer's Responses to Questions

**Comments to the Author**

1. Is the manuscript technically sound, and do the data support the conclusions?

Reviewer #1: Yes

2. Has the statistical analysis been performed appropriately and rigorously? 

Reviewer #1: N/A

3. Have the authors made all data underlying the findings in their manuscript fully available?

Reviewer #1: Yes

4. Is the manuscript presented in an intelligible fashion and written in standard English?

Reviewer #1: Yes

5. Review Comments to the Author

Reviewer #1: This is an interesting and well written paper. It includes important information relating to vector-borne diseases of dogs in North Carolina. I suggest accepting for publication following correcting the below comments.

Major comments

Can the authors explain why cases from 2019-2022/2023 were excluded from the study? I am asking so as 4-5 years, under current global changes, can affect the epidemiologic picture?

L 14-141 – " Primers, PCR conditions, and negative and positive controls were used as previously described (11–14)." Authors should include the list of primers, PCR conditions, and negative and positive controls used at least as a supplemental material. Readers, researchers and reviewers should be able to evaluate the tests' protocols and do not have to look for these important protocols in other articles.

Minor comments

L 109 – should be cells/ul

L 139 – should be Anaplasma spp.

L 170 and legends of Figures 2 & 3 – should be disease (and not diseases)

L 289, 291 and all over the text – spp. should not appear in italics.

Table 4 – lines representing Bartonella spp. PCR and Rickettsia spp. PCR should be removed from the table or the legend should be corrected (as currently includes:"…. dogs testing positive for individual vector borne diseases".

L 414 – Ref 17 – pages should be added.

Supplemental Table 1 – all spp. should NOT appear in italics.

6. PLOS authors have the option to publish the peer review history of their article (what does this mean?). If published, this will include your full peer review and any attached files.

Reviewer #1: No

---

## [Author Response · Author response to Decision Letter 0]

6 Oct 2023

Reviewer's Responses to Questions

Comments to the Author

1. Is the manuscript technically sound, and do the data support the conclusions?

Reviewer #1: Yes

2. Has the statistical analysis been performed appropriately and rigorously? 

Reviewer #1: N/A

3. Have the authors made all data underlying the findings in their manuscript fully available?

Reviewer #1: Yes

4. Is the manuscript presented in an intelligible fashion and written in standard English?

Reviewer #1: Yes

5. Review Comments to the Author

Reviewer #1: This is an interesting and well written paper. It includes important information relating to vector-borne diseases of dogs in North Carolina. I suggest accepting for publication following correcting the below comments.

Major comments

Can the authors explain why cases from 2019-2022/2023 were excluded from the study? I am asking so as 4-5 years, under current global changes, can affect the epidemiologic picture?

We began preparing this retrospective manuscript in 2019, and therefore chose to include cases prior to this date. However, we agree with the reviewer that the epidemiologic picture may change over time, and we intend to re-evaluate the data every 5-10 years to monitor for significant changes. 

L 14-141 – " Primers, PCR conditions, and negative and positive controls were used as previously described (11–14)." Authors should include the list of primers, PCR conditions, and negative and positive controls used at least as a supplemental material. Readers, researchers and reviewers should be able to evaluate the tests' protocols and do not have to look for these important protocols in other articles.

We have included all PCR primers, reaction conditions, and positive/negative controls in a supplemental file (S1 Table). While creating this file, we decided to include several additional references to more comprehensively cite our protocols (see references 11-21). In addition, the following section has been added to lines 142-152 of the materials and methods section: 

“Primers, PCR conditions, and negative and positive controls were used as previously described [11–22]. Briefly, DNA was extracted from stored canine ethylenediaminetetraacetic acid (EDTA) whole blood samples using a commercially available MagAttract DNA blood kit (Qiagen Inc, Chatsworth, CA) according to the manufacturer’s instructions. For all samples, the DNA concentration was quantified by spectrophotometry, and the absence of PCR inhibitors was demonstrated by amplification of a fragment of the glyceraldehyde-3-phosphatase dehydrogenase gene. PCR screening was performed as previously described [11-22]; a comprehensive list of the PCR primers, reaction volumes, and reaction conditions used is available in supplemental table 1. Products were analyzed by 2% agarose gel electrophoresis containing 0.2ug ethidium bromide/mL under ultraviolet light. Canine DNA from a healthy subject was used as a PCR negative control. Positive controls are listed in supplemental table 1.”

Minor comments

L 109 – should be cells/ul

L 139 – should be Anaplasma spp.

L 170 and legends of Figures 2 & 3 – should be disease (and not diseases)

L 289, 291 and all over the text – spp. should not appear in italics.

Table 4 – lines representing Bartonella spp. PCR and Rickettsia spp. PCR should be removed from the table or the legend should be corrected (as currently includes:"…. dogs testing positive for individual vector borne diseases".

L 414 – Ref 17 – pages should be added.

Supplemental Table 1 – all spp. should NOT appear in italics.

We would like to thank the reviewer for catching these errors. All minor comments have been corrected within the revised manuscript. 

6. PLOS authors have the option to publish the peer review history of their article (what does this mean?). If published, this will include your full peer review and any attached files.

Do you want your identity to be public for this peer review? For information about this choice, including consent withdrawal, please see our Privacy Policy.

Reviewer #1: No

---

## [Editor Report · Decision Letter 1]

10 Oct 2023

PONE-D-23-14843R1A retrospective study of vector borne disease prevalence among anemic dogs in North Carolina.PLOS ONE

Dear Dr. Kendall,

Thank you for submitting your manuscript to PLOS ONE. After careful consideration, we feel that it has merit but does not fully meet PLOS ONE’s publication criteria as it currently stands. Therefore, we invite you to submit a revised version of the manuscript that addresses the points raised during the review process.

We look forward to receiving your revised manuscript.

Kind regards,

Joshua Kamani, PhD

Academic Editor

PLOS ONE

Journal Requirements:

Additional Editor Comments:

The authors are invited to kindly address the following comments:

Line 137 Ehrlichia spp add full stop after spp. ‘’.’’ Ehrlichia spp’’.’’ and several places in the manuscript after’’ spp.’’ especially Table 3,4

Line 137 what is Ehrlichia spp ‘’dot’’?

Line 159: Anaplasma spp. not : Anaplasmosis spp. please correct it

Line 183-184 should not be italic.

Line 195. D. immitis instead of Dirofilaria immitis

Line 201. 288. B. canis not Babesia canis’’.’’ Remove ‘’.’’

Line 202, 289. B. gibsoni not Babesia gibsoni

Line 211 correct------2 dogs ‘’with’’ infected with----

Tables 2,3,4. italicize genus and species names appropriately

Line 218 abbreviations should not be italicized

Line 248 remove one ‘’.’’---- anaemia.”.””

Line 303-304-Does this mean dogs examined at the hospital or are they dogs kept at the Institution? Rephrase

Lines 332-334- Recast the sentence appropriately

---

## [Author Response · Author response to Decision Letter 1]

19 Oct 2023

The authors are invited to kindly address the following comments:

Line 137 Ehrlichia spp add full stop after spp. ‘’.’’ Ehrlichia spp’’.’’ and several places in the manuscript after’’ spp.’’ especially Table 3,4 Thank you, this has been edited throughout the manuscript 

Line 137 what is Ehrlichia spp ‘’dot’’? This has been changed to ‘positive spot’ to describe the positive result of Ehrlichia on a SNAP 4Dx 

Line 159: Anaplasma spp. not : Anaplasmosis spp. please correct it This has been corrected. 

Line 183-184 should not be italic. This has been corrected. 

Line 195. D. immitis instead of Dirofilaria immitis Thank you, this has been edited 

Line 201. 288. B. canis not Babesia canis’’.’’ Remove ‘’.’’ This has been corrected. 

Line 202, 289. B. gibsoni not Babesia gibsoni This has been corrected 

Line 211 correct------2 dogs ‘’with’’ infected with---- Thank you, this has been corrected 

Tables 2,3,4. italicize genus and species names appropriately This has been edited in the tables 

Line 218 abbreviations should not be italicized. This is corrected 

Line 248 remove one ‘’.’’---- anaemia.”.”” Thank you, this has been edited 

Line 303-304-Does this mean dogs examined at the hospital or are they dogs kept at the Institution? Rephrase This has been rephrased 

Lines 332-334- Recast the sentence appropriately This sentence has been rephrased

---

## [Editor Report · Decision Letter 2]

23 Oct 2023

A retrospective study of vector borne disease prevalence among anemic dogs in North Carolina.

PONE-D-23-14843R2

Dear Dr. Kendall,

We’re pleased to inform you that your manuscript has been judged scientifically suitable for publication and will be formally accepted for publication once it meets all outstanding technical requirements.

Kind regards,

Joshua Kamani, PhD

Academic Editor

PLOS ONE
---

## [Editor Report · Acceptance letter]

31 Oct 2023

PONE-D-23-14843R2 

A retrospective study of vector borne disease prevalence among anemic dogs in North Carolina 

Dear Dr. Kendall:

I'm pleased to inform you that your manuscript has been deemed suitable for publication in PLOS ONE. Congratulations! Your manuscript is now with our production department. 

Kind regards, 

on behalf of

Dr. Joshua Kamani 

Academic Editor

PLOS ONE